# Generation of Lens Progenitor Cells and Lentoid Bodies from Pluripotent Stem Cells: Novel Tools for Human Lens Development and Ocular Disease Etiology

**DOI:** 10.3390/cells11213516

**Published:** 2022-11-06

**Authors:** Aleš Cvekl, Michael John Camerino

**Affiliations:** 1Departments Ophthalmology and Visual Sciences, Albert Einstein College of Medicine, Bronx, NY 10461, USA; 2Department of Genetics, Albert Einstein College of Medicine, Bronx, NY 10461, USA

**Keywords:** cranial placodes, crystallins, de-nucleation, differentiation, gene expression, lens progenitor cells, lentoid bodies, optic cup, pluripotent stem cells, PAX6, self-organization

## Abstract

In vitro differentiation of human pluripotent stem cells (hPSCs) into specialized tissues and organs represents a powerful approach to gain insight into those cellular and molecular mechanisms regulating human development. Although normal embryonic eye development is a complex process, generation of ocular organoids and specific ocular tissues from pluripotent stem cells has provided invaluable insights into the formation of lineage-committed progenitor cell populations, signal transduction pathways, and self-organization principles. This review provides a comprehensive summary of recent advances in generation of adenohypophyseal, olfactory, and lens placodes, lens progenitor cells and three-dimensional (3D) primitive lenses, “lentoid bodies”, and “micro-lenses”. These cells are produced alone or “community-grown” with other ocular tissues. Lentoid bodies/micro-lenses generated from human patients carrying mutations in crystallin genes demonstrate proof-of-principle that these cells are suitable for mechanistic studies of cataractogenesis. Taken together, current and emerging advanced in vitro differentiation methods pave the road to understand molecular mechanisms of cataract formation caused by the entire spectrum of mutations in DNA-binding regulatory genes, such as PAX6, SOX2, FOXE3, MAF, PITX3, and HSF4, individual crystallins, and other genes such as BFSP1, BFSP2, EPHA2, GJA3, GJA8, LIM2, MIP, and TDRD7 represented in human cataract patients.

## 1. Introduction

The original discovery of embryonic stem (ES) cells and their in vitro cultures triggered new research avenues that transformed both basic and translational research towards better understanding of mammalian embryonic development and human diseases [1,2]. The promise of this field was further boosted by the discovery of nuclear reprograming methods to generate induced pluripotent stem (iPS) cells from mature somatic cells. Over two decades of research using cultured human, monkey, and mouse ES/iPS cells demonstrated their ability to differentiate into a full spectrum of common lineage progenitors and a variety of terminally differentiated cells that exhibit similar morphological, functional, and molecular signatures of individual tissues or even organs [3,4,5,6]. Here, we call organoids in vitro-generated 3D structures from differentiating pluripotent stem cells to individual tissues and organs that exhibit realistic microanatomy compared to their natural counterparts [7]. It has been shown earlier that it is possible to mechanically dissociate cells from chicken organs, such as kidney, liver, and skin, followed by their reconstitution through self-sorting and reaggregation [8]. Experiments conducted during the last 20 years further demonstrated that tissue-progenitor cells can self-organize through cell sorting and exchange of signals into spatially organized mini-tissues and organs [7]. Remarkably, it is now even possible to reconstruct “synthetic” in vitro ~E8.5 mouse embryos derived from three ES-derived cell populations capable to complete gastrulation and early organogenesis [9,10].

Elaborate methods of genome engineering, currently driven by CRISPR-Cas9 technologies [11,12], allow efficient generation of a range of mutations, gene expression reporters, and production of fusion proteins with a variety of fluorescent tags to be tracked even at single-molecule levels [13]. These methods can dissect precise molecular mechanisms specific to any mRNA, ncRNA, and protein of interest. In 2006–2007, the breakthrough discoveries of cellular reprogramming that established mouse- [6] and human-induced [14] pluripotent stem (iPS) cells for the first time now allow routine generations of patient-specific iPS cells carrying defined disease-causing mutations and their corrected version [15,16,17].

Eye development is an intriguing and challenging system to apply pluripotent stem cell technologies to address numerous gaps in our understanding of both the mechanisms of human eye development, eye diseases, and the development of new cell- and drug-based therapies [18,19,20]. Although traditional vertebrate models of eye development established a comprehensive foundation for our understanding of cellular and molecular mechanisms of eye formation, there are several unique features of the human eye (see below) and its underlying gene regulatory networks (GRNs) that can now be modeled using human cells.

Despite these advances, multiple experimental challenges remain to be addressed in the emerging field of 3D eye tissue/organoid engineering. Lessons from normal embryonic eye development continue to serve as invaluable resources for these experiments. In the application stage of these in vitro models, it is important to consider that mutations in individual genes, especially those encoding regulatory proteins, may impact the whole eye, selected ocular tissues, or just an individual tissue, e.g., the lens, cornea, cones, rods, or retinal pigmented epithelium (RPE). Mechanistically, these mutations can act cell-autonomously, non-cell-autonomously, or via a combination of these mechanisms [21,22,23,24,25].

Ocular lens development is a classical model for a highly orchestrated formation of committed lens progenitor cells originating from a transient population of cells at the border between the neuroectoderm and naïve ectoderm, the anterior pre-placodal region [26,27,28]. This inverted U-shape morphogenetic zone gives rise to three types of distinct structures, including a single anterior pituitary placode and more posteriorly, two symmetric pairs of olfactory and lens placodes [27,29] (Figure 1A).

The formation of both lens and olfactory placodes requires temporally regulated BMP signaling [35,36,37]. In addition, FGF signaling in lens progenitor cell formation prevents the formation of the alternate epidermal cell fate [36]. Nevertheless, given the complexity of FGF signaling at the neural plate/neural fold stage [30,38], there is no single or combined gene loss-of-function experiments including individual FGF receptors, FGFs, and other components of this pathway that abrogate lens placode formation. A subsequent reciprocal invagination of the non-neurogenic lens placode and neuroectoderm-derived optic vesicle generates the earliest 3D eye primordium, comprised of the lens vesicle and optic cup [28,30,34]. These eye primordia are surrounded by periocular mesenchymal cells, required subsequently for anterior segment morphogenesis [39,40,41,42,43,44] and covered by the surface ectoderm, giving rise to the corneal epithelium [45]. At its anterior portion, the polarized lens vesicle is patterned by gradients of BMP, FGF, IGF, PDGF, and Wnt growth factors. Spatially regulated expression of these growth factor receptors [46,47,48,49] subsequently forms the monolayer of the epithelium. The posterior portion of the lens vesicle undergoes cell cycle-exit coupled differentiation regulated by complex interactions between BMP, FGF, and PDGF signaling [50,51,52,53] to generate the primary lens fiber cells. The proliferation of epithelial cells generates new cells at the lens equator and their subsequent differentiation generates secondary lens fibers.

The three hallmark features of lens fiber cell differentiation include: (a) robust transcription, translation, and accumulation of crystallin proteins required for lens transparency and light refraction [54,55], (b) formation of elaborate organization of lens membranes and lens fiber cell cytoskeleton coupled to extensive cellular elongation [56,57,58,59,60,61], and (c) highly organized degradation of all subcellular organelles, culminating with lens fiber cell de-nucleation in maturing lens fiber cells, to generate the “organelle-free zone” (OFZ) [62,63,64]. The fully maturated lens fibers exhibit interdigitated “ball-and-socket” junctions [65,66]. The abundant junctions between individual lens fibers implicates that enucleated lens fibers function as a syncytium with respect of free movement of small molecules [67,68]. The lens growth process was quantitatively analyzed in the murine model from the onset of secondary lens fiber cell differentiation to the end of the lifespan [69,70]. These complex processes underlying lens fiber cell maturation [71] must be at least partially recapitulated during in vitro generation of lentoid bodies as well as during the formation of lenses together with retinal tissues.

Why are human eye organoid models derived from differentiated pluripotent stem cells so important? Human eyes evolved far more complex vision and intricate functional anatomy compared to the rodent models used in the laboratory research. Humans employ diurnal vision while rodents rely on rod photoreceptors and nocturnal vision. The human eye contains a unique anatomical structure called the macula and its central portion, the fovea, rich in cone photoreceptors [72,73]. The morphogenetic processes underlying cone-dominant foveal development can be only partially inferred from other model organisms (e.g., 13-line ground squirrels, chameleons, and other reptiles) [74,75,76]. The human eye can accommodate lenses to both near-sight focus and over a distance of several hundreds of meters. Consequently, human lenses are much softer, and two γ-crystallin genes in rodents, Cryge and Crygf (mouse chromosome 1), mutated into a pair of human pseudogenes located on the syntenic region of human chromosome 2 [34] likely contributing to this phenomenon through the overall reduction of γ-crystallin proteins in the lens nucleus. Finally, in vitro generation of artificial human tissues can address the scarcity of authentic ocular tissues, allowing for expansion of more in-depth human biochemical studies and other analyses.

Here, we examine the formation of lens progenitor cells and subsequent generation of lentoid bodies to study human lens development and model cataract formation and lens pathology. We organize different experimental procedures into groups with shared features and outcomes. In Section 2, we include a pioneering study using mouse ES cells to generate whole eyes published in 2003. Section 3 is focused on eye-independent formation of lentoid bodies and 3D-formation of optic cups to illustrate the general self-organization principles introduced above. Induction of cranial placodes in multiple systems is used to illustrate the function of regulatory genes and signaling pathways required for normal lens morphogenesis (Section 4). Advanced systems to generate 3D lentoid bodies based on cell-sorting principles are included in Section 5. In Section 6, we discuss modeling of cataract and other human ocular diseases using pluripotent cell differentiation, genome engineering, and application of unbiased multi-omics methods for analyses. Finally, Section 7 provides a summary of current challenges in tissue engineering to form ideal lenses and eye-like structures and the future direction of this fascinating research.

## 2. Prologue (2002–2003)

Mouse ES cells were first derived from the inner cell mass of the early embryos in 1981 [1,77]. Subsequently, generation and initial characterization of human ES cells followed in 1998 [2]. Early pioneering studies have shown differentiation of human and mouse ES cells into a representative number of mature cells such as dopamine neurons [78], motor neurons [79], and RPE cells [80].

Early research on lens differentiation has demonstrated that in vitro-cultured lens epithelial cells can be differentiated into primitive 3D lens-like structures, termed “lentoid bodies” [81,82,83,84,85,86,87,88,89,90,91]. Lentoid bodies are comprised from aggregated cells all expressing crystallin proteins and some cells exhibit elongated morphology. Importantly, some of these studies directly showed that lentoid bodies are both transparent and refract light. Hence, these early studies supported the general concept of self-organization [7,92] in 3D-cultures (see Section 6 for details) as a foundation of contemporary tissue engineering.

Interestingly, lentoid body-like structures were also found in vivo in embryos of diverse model vertebrates as the result of spontaneous or engineered mutations in genes involved in lens morphogenesis [93,94,95,96,97] or via ectopic expression of the lens regulatory gene Six3 in medaka fish [98] and Pax6 in frogs [99]. An elegant study has shown that lens formation can be restored in Rax-depleted mouse embryos through the elimination of β-catenin expression, a major player of the canonical Wnt signaling, in the head surface ectoderm [100]. Trans-differentiation of RPE cells into lentoid bodies also illustrates that primitive lens formation is not required to be inside of the optic cup [81,101,102]. Lens regeneration from the dorsal iris in newts is another example of developmental plasticity [103]. Taken together, these studies demonstrated lens formation under various abnormal conditions. It was just a matter of time to detect lenses in early attempts to “spontaneously” differentiate pluripotent stem cells into various ectoderm-derived tissues.

The first pioneering studies of in vitro formation of lens and retinal cells from mouse pluripotent stem cells indeed generated remarkable outputs [104,105]. The first system used 129/SV mouse ES cells in the absence of any feeder cells and leukemia inhibitory factor to generate embryonic bodies [104]. A subsequent treatment with 5 and 50 nM of retinoic acid (RA) promoted formation of the optic vesicle, lens, and retina, as well as other cell types in 20 days of cultures; however, these cells were only visualized via rudimentary histology and without detection of any cell-specific protein markers.

The other system employed the PA6 stromal cells derived from C57/Bl6 mice as a feeder layer required for differentiation as they produce neural-inducing factors [106,107]. The aggregated colonies of cells were observed at day seven of cultures, and, two days later, globular structures including weakly pigmented RPE-like cells were detected [105]. More complex eye-like structures, including primitive lenses validated by studies of crystallin gene expression, were found as early as at day 11, around the time when mid-gestation mouse embryos generate the lens vesicles and early optic cups. Addition of bFGF/FGF2 further improved induction of eye-like structures. The presence of PA6 feeder cells was critical and the use of bFGF could not replace them. Finally, this study demonstrated that the use of Pax6−/− ES cells did not produce any of these structures using nearly 3000 early cell colonies generated from the mutated cells. As expected, Pax6+/− ES cells retained this ability [105]. Taken together, this single study demonstrated for the first time a “proof-of-principle” that complex 3D eye structures can be generated in vitro without any sophisticated procedures and unusual time constraints.

In parallel, cynomolgus monkey ES cells were grown on PA6 stromal cells as a source of differentiation factors plated on gelatin-coated dishes [108]. Lentoid bodies were detected from day 14–16 of cultures and analyzed up to day 53. FGF2 was tested at concentrations of 2, 4, and 8 ng/mL and improved the yield of lentoid bodies [108]. It is of interest that the “parental” protocol generates pigmented epithelial cells and dopaminergic neurons [109].

Nevertheless, these three initial studies were underappreciated as iPS cell technologies emerged later in 2006. The generation of iPS cells via cellular reprogramming was a game changer in the field of pluripotent stem cell differentiation [6] as it bypassed the use of human early embryonic materials, facilitated generation of iPS cells from both normal individuals and patients with specific genetic abnormalities, leading to diseases [14], allows correction of the genetic defect(s), and facilitates large-scale generation of interesting mutations using isogenic iPS cells (see Section 6.2 for details). In addition, dynamic visualization of the formation of the optic cup using fluorescent proteins was also employed much later in 2011 [110,111]. Finally, in 2001, another important study showed that Matrigel can replace feeder cells to grow and expand ES cells [112]. The extracellular matrix known as Matrigel is isolated from mouse Engelbreth–Holm–Swarm teratocarcinoma cells that secrete excess basement membrane-type extracellular matrix (ECM) instead of cartilaginous matrix proteins [113]. Biochemical data show that Matrigel is a gelatinous mixture of collagen IV, laminin, proteoglycans, entactin, and other growth factors [114] that also promotes differentiation of various cell types [115], including lens cells [116].

## 3. Independent Formation of Lentoid Bodies and Optic Cups (2010–2016)

A specific goal to generate in vitro lens progenitor cells and lentoid bodies has been achieved primarily through lessons taken from lens developmental biology [35,36,46,117,118,119,120], including the role of ECM proteins [116,121,122,123,124,125]. The Matrigel-based three-stage procedure (Figure 2A) is based on neuroectoderm formation via Noggin (stage 1).

Stage 1 was followed by activation of BMP and FGF signaling by BMP4, BMP7, and FGF2 to generate lens progenitor cells (stage 2) and completed by FGF2 with or without Wnt3a stimulation of lens fiber cell differentiation to generate 3D lentoid bodies (stage 3) using human H1 ES cells (WiCell) [126]. Matrigel (see above) was particularly selected as the major lens capsule core structural proteins are laminin [122] and collagen IV [121], together with smaller quantities of nidogen [123] and perlecan [125]. Lens progenitor cells were identified via expression of transcription factors PAX6, SIX3, and SOX2 and the onset of αB-crystallin (CRYAB) expression (Figure 2B). Lentoid body formation peaked by day 29, producing over 100 lentoid bodies per cm^2^. The lentoid bodies (day 35) were analyzed by scanning and transmission electron microscopy, immunofluorescence, and Western immunoblotting. The cells forming lentoid bodies expressed high levels of crystallins; however, enucleated lens fibers were rarely detected at day 35 of the cultures [126]. This timing could be a consequence of the fact that normal lens fiber cell de-nucleation in humans occurs much later, as primary lens fibers are formed at seven weeks of gestation. Taken together, this three-stage protocol from 2010 recapitulates three phases of normal lens formation, i.e., formation of neuroectoderm and surrounding naïve ectoderm, emergence of lens progenitor cells, and their transition into crystallin protein-expressing cells.

Using this protocol, age-related cataract patients were used as a source of lens epithelial cells to be reprogrammed into iPS cells via lentiviral transduction of Oct4, Sox2, Klf4, and c-Myc “Yamanaka factors” [6]. The individual human iPS cells were subsequently used to remake the “normal” lens cells and lentoid bodies [127,128] using the three-step protocol [126].

A different approach was based on the ability of Pax6 and Six3 to induce ectopic lenses (see above). Pax6 or Six3 expression vectors were used to transduce human and mouse ES cells grown on inactivated mouse embryonic fibroblast feeders, including engineered mouse cells expressing green fluorescent protein under the control of the Pax6 lens-specific promoter/enhancer in 2014 [129]. Interestingly, lens markers were expressed by cells close to the transduced cells, suggesting that these cells adopt a lens fate by a non-cell-autonomous mechanism [129]. Cell aggregates expressing γA-cystallin proteins were already found at day 7 following their transition into lentoid bodies analyzed at day 30. Both the lens epithelial cell marker transcription factor Foxe3 and fiber cell abundant transcription factor Prox1, RNA-binding protein Tdrd7, α- and β-crystallins were expressed in the mouse ES cell-derived lentoid bodies [129].

In another study, mouse iPS cells from fetal fibroblasts were generated from transgenic mice in which the αA-crystallin promoter [130] was linked to the tdTomato fluorescent reporter gene [131]. The non-viral reprogramming was conducted using a “Sleeping Beauty” multi-cistronic transposon system including mouse Oct4, Sox2, Klf4, and c-Myc cDNAs separated by sequences coding for the self-cleaving 2A peptides [132]. Three cell lines (NTERA-2, P19, and control STO cells) were evaluated as feeder cells, while no individual growth factors were tested. Human NTERA-2 cells are committed neuroectodermal cells and mouse P19 cells are known to differentiate into all three germ layers. The first tdTomato-positive cells were found around day 28 of cultures using NTERA-2 and P19 feeder cells. Some of these cells organized themselves into lentoid bodies analyzed at day 45 for light refraction [131]. Note that normal mouse lens morphogenesis, defined by the formation of OFZ, is completed around E18.5. Thus, the feeder cells likely express suboptimal concentrations and/or combinations of factors that promote lens cell fates and lentoid bodies.

Three of the four mentioned studies [126,129,131] have independently concluded that relatively simple cell culture conditions generate 3D lentoid bodies analyzed in detail between 5 and 7 weeks of cultures. Engineering of cells with fluorescent reporters provides advantages for rapid screening of the differentiating cells [129,131]. Nevertheless, no systematic analyses of cell fate decisions towards the lens and its alternatives as well unbiased analysis of all cell populations present on the dish and their dynamic changes during the individual stages of the three-stage protocol [126,127] or both single-stage protocols were conducted [129,131].

Regarding contemporary eye research (2011–2012), parallel studies generated 3D optic cups from ES/iPS cells using Matrigel and a few distinct small-molecule drugs without any lenses within their concave openings [110,133]. These studies highlighted the concept of self-organization (see Section 1, Section 5.2 and Section 7 for additional information) as the mechanistic foundation explaining in vitro formation of 3D biological structures [7]. For comparisons, the procedures to generate retinal and eye organoids of various complexity [105,110,111,126,129,131,134,135,136,137,138,139,140,141,142,143,144,145,146,147,148,149,150,151,152,153,154,155,156,157,158,159,160,161,162] are listed in Table 1, as they are important for human eye disease modeling (see Section 6).

## 4. Complex Differentiation Procedures: Induction of Cranial Placodes and Parallel Formation of Lenses Together with Other Ocular Tissues

Although generation of isolated lentoid bodies alone is an important model system with inherited limitations, some of which are described above, production of lens cells together with other ocular tissues (“community-grown” lenses) should provide additional opportunities for research focused on mechanistic aspects of early lens development and eye tissue engineering. This section will thus provide background and accomplishments of the field to generate in vitro adenohypophyseal, olfactory, and lens anterior cranial placodes, normally produced during late stages of mammalian gastrulation.

Early lens development originates from cells located at the border between the anterior neural plate and surface ectoderm, called the anterior pre-placodal region [27,28,33,34,163,164,165]. The common Pax6^+^ cells emerging in the pre-placodal region give rise to three cranial placodes, including the non-neurogenic adenohypophyseal/anterior pituitary placode, neurogenic olfactory, and non-neurogenic lens placodes (Figure 1B–D). Thus, it was of interest for both the lens research field and the basic science questions regarding growth factor-directed differentiation of ES and iPS cells towards neuroectoderm [166] to directly examine the formation of all early placodal progenitors [140,141,142,143]. Other studies probed formation of more posterior optic placodes [167] and SOX10^+^ neural crest cells [142,168,169].

### 4.1. Dual-SMAD Inhibition and Generation of Placodal Precursors

It has been shown earlier that neuronal cells are induced via concomitant inhibition of BMP and TGF-β, Activin, and Nodal signaling by Noggin and small drug SB431542 (see Table 3), called dual-SMAD inhibition (dSMADi), using Matrigel-coated dishes [166]. De-repression of endogenous BMP signaling that follows the initial generation of neuronal cells, typically after day 3 of cultures (Figure 3), results in robust generation of over 70% of placodal precursor cells marked by expression of SIX1, DLX3, and EYA1 (Table 2) by day 11 of the cultures [140].

Regarding lens placodal cell fate (Table 2), re-analysis of data shows increased expressions of transcription factors GATA3 [170] and TFAP2A [171] at days 7 and 11 of the cultures, respectively. At day 11, the cultures also increased the expression of ALDH1A3 and ALDH1A1, both found in the early mouse pre-placodal cells (Table 2). Use of the specific inhibitor of FGF signaling SU5402 (Table 3) blocked placodal cell formation and promoted the formation of surface ectoderm marked by high levels of keratin 14 (KRT14) expression (Figure 3). Increased concentrations of BMP4 also suppressed placodal cells while formation of putative trophectoderm cells increased.

Four cell lineages were characterized in detail in this pioneering study, including anterior pituitary, lens placodes, and neurogenic trigeminal sensory neuron placodes (not shown in Figure 3), as well as the ectoderm pathway. Purified placodal cells can be subsequently differentiated by a combinatorial action of Sonic hedgehog (SHH), FGF8, and FGF10 into pituitary-like cells at day 30 of the cultures [140], and subsequently into three groups of hormone-secreting pituitary cells by FGF8 or BMP2 alone, or by a combination of FGF8 and BMP2 [141]. The procedure did not show any data on the olfactory placode; nevertheless, the authors stated that FGF8 increased ASCL1 [140]. In fact, multiple genes found in the olfactory placodes (Table 2), including ALDH1A1, GATA3, FOXG1, and SIX1, were induced between days 5 and 11, as shown by their transcriptomic and/or protein analyses. Addition of BMP4 at days 7–11 increased the expression of lens lineage-specific transcription factors PAX6 and PITX3. Surprisingly, use of SU5042 (see Table 3) at days 7–11 of cultures also promoted expression of both these genes, culminating in detection of αB-crystallin (CRYAB)-positive cells at day 57 (Figure 3).

### 4.2. Alternate Methods to Generate Multiple Placodal Progenitors

Two additional independent studies also examined the formation of multiple placodal progenitors, including lens cells (Table 1), using human H9 ES cells grown on Matrigel [143,144]. The undifferentiated human ES cells were first dissociated into single cells and grown on Matrigel in the presence of standard human ES cell medium supplemented with 8 ng/mL of FGF2 and Rho kinase inhibitor Y27632, that was removed 24 h after the passage [144]. After day 2 or 3, three different media were used to direct differentiation towards neural and neural crest, pre-placodal ectoderm, and epidermis. Serum-free medium promoted the expression of both early (SIX1, GATA3, and TFAP2A) and advanced (SIX1, EYA1, EYA2, and DACH1) pre-placodal markers (Table 2), regulators of lineage-specific transcription. Addition of 20 ng/mL of BMP4 promoted placodal formation. In contrast, Noggin (300 ng/mL) reduced the expression of these markers and the number of SIX1^+^ cells [144]. Interestingly, transitional addition of the drug inhibitor of BMP signaling LDN193189 (Table 3) increased the expression of PAX6 and the number of PAX6^+^ cells increased 7-fold, with 90% of them co-expressing both PAX6 and SIX1 and PAX6 and DLX5 transcription factors. Prolonged cultures in serum-free conditions expressed high levels of PAX6 and FOXE3 but no αA-crystallin (CRYAA) expression was found. Addition of 20 ng/mL of BMP4 induced the expression of αA-crystallin proteins [144]. Next, bovine vitreous humor was used as a source of natural factors, promoting lens differentiation [185]. While primitive spheres of lens cells were found, no signs of lens fiber cell differentiation were observed, pointing to differences between in vitro-generated lens cells and primary lens cell cultures. The data further showed expression of two corneal markers, keratins KRT3 and KRT15, implicating induction of corneal epithelial cells. Finally, SHH signaling was examined via the inhibitor cyclopamine (Table 3) and the SMO receptor agonist purmorphamine. As expected from earlier studies [41,186], inhibition of SHH increased the expression of FOXE3 and CRYAA, while addition of purmorphamine abolished the expression of lens markers and increased the expression of oral ectoderm marker transcription factors PITX1 and PITX3 [144].

A parallel study of the formation of cranial placodes from the neural plate border employed fluorescence-activated cell sorting (FACS) to analyze proportions of lens cells within the cultures [143]. For the positive selection of lens cells grown in serum-free medium (Table 1), the hepatocyte growth factor receptor (HGFR, also known as c-MET) was used, while p75, HNK1, and CD15 were used for the negative selection. More advanced cell cultures showed expression of the hyalouran receptor CD44, which is expressed in cortical lens fibers [187]. Quantitative analyses showed very small percentages (<1%) of putative lens cells as both these receptors are also expressed in many different cell types. The small fraction of p75^−^, HGFR^+^, HNK^−^, CD44^+^, and CD15^−^ cells generated lentoid bodies within 10 days, analyzed by the expression of CRYAA, BFSP1, and MIP genes [143].

### 4.3. Placodal Cells and Their Quantities

The three studies discussed above demonstrate that cell fate choices in differentiated pluripotent stem cells generally recapitulate normal embryonic development; however, their specific outcome is dictated by an intricate complexity of cell fate decisions governed by temporally and spatially controlled canonical signal transduction pathways, in the case of anterior neuronal development and its nearby border region, the BMP, FGF, and SHH signaling pathways [188]. While olfactory placode formation is a major component of these processes during normal embryogenesis (Figure 1), the above studies provided very little insights into this pathway. Nevertheless, a recent study employed human iPS cells (WiCell) grown on Matrigel using serum-free ITS medium (see above), which generates olfactory placodal cells [189]. Inhibition of BMP signaling via LDN193189 (Table 3) promoted the expression of genes encoding transcription factors DLX5, EMX2, and POU2F1 and blocking the expression of FOXE3. Additional growth factors tested for olfactory placodal cells included FGF8, TGF-α, and SB431542, and Wnt signaling inhibitors [189]. Taken together, lens cells are generated by these “universal” systems at much smaller quantities (<1%) compared to the 41% of PAX6^+^ and CRYAA^+^ cells generated by the 3-stage protocol [126]. A possible interpretation of these efficiencies may be related to the model in which lens cell fate specification is the ground state for other anterior placodes [190].

## 5. Advanced Procedures to Generate Lentoid Bodies

### 5.1. Cell Sorting and Isolation Prior to Cell Differentiation

Two advanced methods further explored efficient generation of lentoid bodies via manual dissection of cell aggregates [136] or cell surface-driven isolation of lens progenitor cells [137]. Using the three-stage lens differentiation protocol [126], the mechanical isolation of cell aggregates both increased the efficiency and reduced the time taken for the formation lentoid bodies. This 2017 protocol was coined as the “fried egg” method due to the selection of colonies that formed dense clusters of differentiating cells surrounded by support cells. These “fried egg”-like colonies were characterized as a dense center of E-cadherin-positive cells surrounded by a monolayer of E-cadherin-negative cells, where differentiation of lentoid bodies would occur from the center.

After Noggin treatment for six days, cells with epithelial-like morphology around the periphery of the colonies were mechanically isolated, re-seeded, and cultured under the three-stage differentiation conditions. By day 11, colonies that have formed “fried egg”-like structures were further differentiated. At day 14, immature lentoid bodies began to form. These immature lentoid bodies were characterized by loss of expression of the lens epithelial marker FOXE3 when compared to the surrounding support cells. Cells were then differentiated in the presence of FGF2 and WNT3A. By day 25, mature lentoid bodies expressed αA-, αB-, β-, and γ-crystallin proteins [136].

An important finding during the development of this method was the effect of cell-seeding density and colony size on the efficiency of lentoid body formation. Optimal seeding density was ~40 colonies per 35 mm culture dish containing ~50 cells per colony, which yielded the largest size and number of lentoid bodies. Higher seeding densities either reduced the number of “fried egg” colony formations or created colonies that had “multiple fried egg” structures that did not produce lentoid bodies [136]. Transcriptome profiling of lentoid bodies generated from the “fried egg” method show a >96% overlap in differentially expressed genes (DEGs) when comparing iPSC- vs. hESC-derived culture systems with a 95% quantitatively similar transcriptome profile (within 2 standard deviations) [136].

Importantly, proteome changes of lentoid bodies generated from the “fried egg” using both 25- and 35-day cultures revealed little to no change in proteome when compared to P0.5 mouse lens epithelial and fiber cell proteomes [136], reaching concordance levels of >93% and >92%, respectively [191]. When compared with E14.5 mouse lens epithelial and fiber proteomes, 25- and 35-day cultures showed 89% and 90% overlapping of lens proteins, respectively [192]. More than 5000 proteins not reported earlier [193] were expressed in lentoid bodies [191]. Hence, the authors proposed that non-lens-associated proteins are from a mixed population of lens-associated cells that support lentoid body differentiation and maturation [191].

In a parallel study, cell sorting of lens epithelial cells based on the expression of the receptor tyrosine kinase-like orphan receptor 1 (ROR1) surface marker was accomplished in 2018 [137]. ROR1 was identified from previous lens microarray data as a potential cell surface marker for lens epithelial cells [194]. Further analyses found enriched expression of ROR1 in E14 mouse lenses and during the second stage of the original three-stage procedure. In addition to the magnetic-activated sorting (MACS) step, this method increased the Noggin concentration from 100 to 500 ng/mL and included 10 mM of SB431542 (Table 3) in stage 1 (Figure 2). Stage 3 was also modified by reducing the concentration of FGF2 to 10 ng/mL. Collectively, these conditions optimized the previous “naïve” protocol via selective proliferation of the ROR1^+^ cell population [137].

Hence, for the first time, this modified protocol allows for the large-scale purification of lens epithelial cells from human pluripotent stem cells [137]. Prior to this method, lentoid bodies were generated spontaneously with random shape, size, and a heterogeneous population of cells. This process was scaled using forced aggregation of ROR1^+^ cells into ~100 µm aggregates at a density of 1200 aggregates per well of a 24-well plate. Cell aggregates were cultured for three weeks in the presence of both FGF2 (100 ng/mL) and WNT3A (20 ng/mL). After only three weeks, lentoid bodies exhibited a remarkable refractive and light-focusing capability. Interestingly, the light transmittance quality of lentoid bodies was associated with changes in α-, β-, and γ-crystallin expression levels. Particularly, α- and β-crystallins were most abundant in lentoid bodies that focused light. Bulk RNA-seq data and analysis of DEGs also revealed expression of lens-specific integrins, laminins, and collagens that indicate the formation of the lens capsule. Evidence of lens capsule formation was also seen under ultra-structural analysis via transmission electron microscopy [137]. This method can generate tens-of-millions of human lens epithelial-like cells [139]. The analyses included unbiased single-cell RNA-seq (scRNA-seq) and mass spectrometry combined with light and electron microscopy, including microscopical evidence for the formation of gap junctions between fiber-like cells, general reduction of subcellular organelles, and early stages of de-nucleation. Additional analysis of the scRNA-seq data obtained from ROR1^+^ lens epithelial cells was reported elsewhere [195].

Taken together, concerted efforts from multiple directions established diverse experimental models to generate lens progenitor cells at different efficiencies. High-yielding lens systems have the power of generating 3D lentoid bodies and micro-lenses of different complexity, while low-yielding systems of lens progenitor cell formation are highly relevant to study cell fate decisions that mimic the formation of the anterior pre-placodal ectoderm and cell signaling processes underlying the formation of non-neurogenic adenohypophyseal and lens and neurogenic olfactory placodes.

### 5.2. Complex Cell Cultures Generating “Byproduct” Lens Cells

Three-dimensional-laminated neural retinal organoids can be produced from human H9 ES cells in a minimal medium containing neural supplements B27 and N2 [196]. Low-attachment bacteriological dishes were used. A follow-up study in 2015 found that addition of IGF-1 (Table 3) at 5 ng/mL until day 37 and at 10 ng/mL in basal knockout serum-free medium (until day 90) stimulates production of bi-layered optic cups with maturing photoreceptor cells, though most of the organoids had reverse laminar organization (Table 1) [179]. Interestingly, clusters of lens-like cells were found in these cultures and immunofluorescence data exist for lens proteins PAX6, SOX1, CRYAA, and CRYAB. Some of these cells did not generate positive nuclear staining, indicating active de-nucleation processes. In addition, emergence of corneal epithelium-like cells was visualized by keratin 19 (CK19) [179].

Another system, called “self-formed ectodermal autonomous multi-zone” (SEAM), was developed in 2016 from a collection of human iPS cells from the RIKEN Bio Source Center using StemFit medium and LN511E8 isoform laminin-coated dishes (Table 1) [134]. These cells spontaneously form multiple primordia comprised of four concentric zones, 1–4, that were characterized by a collection of specific protein markers to identify different cell types in analogy with the human eyeball. For example, PAX6 was expressed in the inner zones 1–3, while the most outer zones 3 and 4 expressed surface ectoderm transcription factor TP63 and epithelial surface E-cadherin [134]. In the most inner zone 1, neural cell-specific transcription factors SOX2 and SOX6 and β3-tubulin (TUBB3) are expressed. Zone 2 expressed optic vesicle and neural crest cell transcription factors RAX and SOX10, respectively. At the margin of zones 2 and 3, αA-crystallin (CRYAA)-positive lens-like cells formed after four weeks of culture. Zone 3 is comprised from the future cells forming the ocular surface epithelium marked by PAX6, TP63, E-cadherin, and keratins K14 and K18 [134]. It was shown later that the 211 isoform of the E8 fragment of laminin (LN211E8) used as ECM promotes generation of dense hiPSC cell colonies due to actomyosin contraction. In turn, this leads to cell density-dependent YAP inactivation and subsequent retinal differentiation in the colony centers [197].

Consistent with earlier studies, the authors found that BMP signaling inhibitors (Table 3) Noggin and small-molecule LDN193189 block the formation of zone 3 and SB431542 disrupts the standard multi-zone formation, respectively [134]. No additional data on lens cells are available as the follow-up experiments focused on modeling of corneal development; nevertheless, several possible treatments and cell enrichments exist based on the procedures summarized above. Both individual treatments and a range of concentrations, and their combinations, should be empirically tested during and after the basic 28-day SEAM protocol.

Finally, a two-step procedure to obtain self-organized multizone ocular progenitor cells (mzOPCs) on Matrigel was optimized to form 3D retinal, corneal, RPE, and multi-ocular organoids (Table 1) in 2021 [135]. While solid progress was accomplished towards these systems, lens cell formation was documented only via sparse detection of γ-crystallin-positive “lentoid cell clusters”.

### 5.3. General Lessons from In Vitro Lens Cell Formation Studies in Diverse Pluripotent Cell Cultures

The lens-generating differentiation procedures used a wide range of experimental conditions (see Table 1, top nine procedures). Matrigel and a laminin variant (LN211E8) were employed as ECM proteins, and PA6, NTERA-2, P19, and embryonic fibroblast cells were used as feeders and/or inducers of differentiation (see above). The emerging common denominator between these procedures is the generation of neuroectodermal cells that are most likely a default stage of the normal ectoderm [198]. Termination of Noggin and dSMADi “deterministic” stages allow the cultured cells to exit this default mode and promote the emergence of new common progenitor lineages, including anterior placode progenitors. The subsequent steps of individual cultures differ from no growth factors/drugs added, to single (e.g., BMP4, IGF-1, and SU5402) and multiple combinatorial treatments (e.g., BMP4, BMP7, and FGF2, see Figure 2). Although no procedure quantitatively analyzed parallel formation of lens cells and other cell types, their variable outcomes reflect the heterogeneity of the presumptive transitional populations of biased, specified, and determined cell types and their responses to activation of TGF-β/BMP [48] and activation or repression of FGF/MAPK [49] signal transduction pathways. We propose that variable expression of multiple individual BMP and TGFβ type 1 and type 2 receptors and FGFR1-4 dimers and their particular sensitivities to the growth factors and/or their antagonists or agonists present on the cell surface modulate the outcomes towards each desired cell type, i.e., adenohypophyseal, olfactory, and lens placodes, as well as more posterior otic placodes [167]. In any case, the individual specified and determined placodal cells undergo migration and spontaneously self-organize into cell-homogenous clusters, from which more complex structures, such as the lentoid bodies/micro-lenses, ultimately emerge.

These cell culture models should help to resolve remaining questions regarding the cell fate determination of the individual placodal cells [188]. Chicken embryonic cell manipulations suggest that the lens program dominates early following placodal progenitor cell diversification [190], while other studies support the model of common lens/olfactory progenitors [36]. A paradox exists between the role of FGF2 and the SU5402 inhibitor of FGFR1 (Table 3) to generate lens cells in the three-stage [126,136,137] and dSMADi [140] protocols, respectively. Is it possible that the expression levels of individual FGFR2, FGFR3, and FGFR4 receptor proteins and their affinities to the prevalent FGF1, FGF2, FGF3, FGF8, FGF15, and FGF24 compensate for the selective block of FGFR1? Both these questions can be addressed via scRNA-sequencing and identification of transient populations of cells and determining their developmental trajectories and specific cell surface receptors, cell adhesion molecules, and ECM proteins required for their self-organization [199].

Finally, four treatments in combination with other growth factors offer improvements for the generation of lentoid bodies described above (see Section 3, Section 4 and Section 5). Early studies of IGF-1 [124,200] suggest use of this growth factor (Table 3) beyond what has been already accomplished [179]. In addition, RA is a powerful morphogen during normal lens development, crystallin gene expression, and lens regeneration [104,201,202,203,204,205,206]. Early studies in 1988 have shown that in vitro pulsatile delivery of platelet-derived growth factor (PDGF) dramatically increased rat lens growth in serum-free conditions [207]. PDGF is a dimeric glycoprotein and a potent mitogen binding a tyrosine kinase cell surface receptor PDGFR and cross-talks with both FGF/MAPK and phosphatidylinositol 3-kinase (PI3K) signaling [53]. PI3K signaling is necessary for avian lens fiber cell differentiation and survival [208]. Importantly, a recent 2022 study of drug-mediated inhibition of PI3K signaling demonstrated that this pathway activates autophagy and generation of lens OFZ [209].

## 6. 3D-Eye Organoids: Experimental Challenges and In Vitro Modeling of Human Ocular Diseases

Significant progress has been made in the generation and molecular characterization of diverse systems, leading to the formation of lens progenitor cells and both isolated and “community-grown” lentoid bodies, as described above. Nevertheless, this field is poised for major advancements to produce the “next generation” of lens and eye organoids to more precisely model human eye development and diseases, provided that more accurate experimental conditions are established. To achieve these goals, we need to consider unique features of normal lens morphogenesis and current bioengineering tools.

During the formation of the lens vesicle and optic cup, the posterior portion of the lens interacts with a transient hyaloid vascular system, leaving the optic cup via its transient opening, the optic cup fissure [28,30,210,211,212]. This system, called tunica vasculosa lentis, is regulated by VEGF-A [213,214], transcriptional co-activator Cited2 [215], and cyclin-dependent kinase inhibitor 2a, Cdkn2a (other names: Arf and p16Ink4a) [216,217]. During the formation of primary lens fibers and closure of the optic fissure (E13.5–E14.5), this vascular system retrogresses and disappears, and the cavity between the lens and the retina is filled by the transparent vitreous humor gel [218]. Consequently, the anterior aqueous and posterior humors are the only source of oxygen for the lens [219,220,221,222]. Low levels of oxygen contained in the aqueous humor combined with the consumption of oxygen by the mitochondria-containing lens epithelial cells at the surface of the lens contribute to the overall low-oxygen environment at the lens surface [221,222]. Studies on interior lens oxygen levels revealed a 20-fold decrease in oxygen levels from the anterior surface of the lens to the equatorial region [221]. Consistent with this hypoxic environment, depletion of the basic helix-loop-helix transcription factor Hif1α in the mouse lens results in disrupted lens growth and ultimate degeneration [223]. Hif1α is the key regulator allowing cells to function in hypoxic conditions via blocking their ubiquitination [224,225]. When stabilized in hypoxia, its target genes promote survival in low-oxygen conditions. Most recent lens studies provide additional insights into these processes via detailed studies of Hif1α and batteries of its regulated genes, including direct regulation of mitophagy regulatory gene encoding BCl2-interacting protein-3-like protein Bnip3l by Hif1α [64,226,227].

Lens physiology is dramatically affected by degradation of mitochondria via mitophagy regulated by BCL2-interacting protein 3-like (Bnip3l, Nix) [228] as a prerequisite of the OFZ formation [62,63,229]. Thus, the majority of the lens requires anaerobic glycolysis, while oxidative phosphorylation is limited to outer regions of the lens fiber cell compartment [230,231]. In addition, the posterior vitreous contains a lower concentration of oxygen compared to the anterior aqueous humor chamber [232]. Finally, it is well-established that normal lens development and postnatal growth are regulated predominantly by BMP, FGF/MAPK, and Wnt signaling pathways [28,46,48,49,233]. Evidence exists that individual growth factors and their receptors are not evenly expressed along the lens vesicle and optic cup [27,35,234,235,236,237]. Taken together, lens physiology evolved very complex temporal and spatial regulation of key processes linked to its precise metabolic requirements as oxidative stress-caused damage to the lens proteins is detrimental to its transparency [238,239,240,241,242]. Thus, lens physiology under hypoxia represents an important parameter to be considered for the next generation of lentoid bodies.

### 6.1. Tissue Engineering and “Next-Generation” Eye Organoids

The next-generation lentoid bodies can be both grown isolated and as “community-grown” structures. Although current advanced procedures generate isolated lentoid bodies with some enucleated cells [136,137], these procedures are performed under normal oxygen concentrations. Thus, the next logical step is to perform “late” stages of their differentiation in incubators, mimicking hypoxic conditions, testing a gradual reduction within ~10 days of culture or a stepwise reduction to 2–3 different lower concentrations, followed by their detailed morphological analyses. However, these procedures may not be suitable for community-grown lentoid bodies as other tissues require a normal oxygen concentration and some moderate hypoxia gradients are generated if the lens-like structures are surrounded by a mass of retinal cells.

The advanced differentiating lentoid bodies, classified by their size, shape, and transparency, can be transferred into Matrigel wells with variable levels (e.g., 1–5 mm levels of Matrigel) or scaffolds from various neutral materials (agarose) that contain a cluster of micropatterned elliptically shaped wells to support further growth of individual “reseeded” lentoid bodies. Various multi-compartmental biomaterial scaffolds are being developed and tested for 3D-cultures of pluripotent stem cells [243,244]. Physical properties such as biomaterial stiffness, crosslinking capacity, and density can aid cellular differentiation, especially if individual growth and differentiation factors are supplied with the biomaterials [244,245,246]. A more advanced approach is to design scaffolds that contain 50–100 ng/mL of FGF2 and the cell culture medium will contain only lower FGF2 concentrations (e.g., 5 ng/mL) to support growth and survival of the lens epithelium. A higher FGF2 concentration at the bottom of the well should promote lens fiber cell differentiation [247]. Gradients of BMP2 and FGF2 were already used in sub-micron polystyrene fibers and evaluated in myocyte differentiation [248]. More recent examples are printed collagen/chitosan scaffolds with stable FGF2-STAB [249]. An alternative to Matrigel is to prepare proteins from bovine lens capsules at a large scale or to model capsule protein composition using ECM proteins collagen IV, laminin, and smaller amounts of nidogen and perlecan [250,251]. Additional proteins to include are various heparane sulfate proteoglycans as they bind FGF1 and FGF2 [252,253] as well as entactin [254].

Another possibility is to mechanically insert lentoid bodies into the optic cups, initially grown individually. Most likely, the optimal approach would be to print “elastic” scaffolds to mimic the optic cup shape and transfer optic cups from floating cultures and allow them appropriate time to adjust to these microenvironments. Following their adaptation, individual lentoid bodies would be “transplanted” and covered by a biodegradable sheet to assure their smooth adoptions by the individual optic cups.

The original method to produce whole eyes [105] can also be tested using human pluripotent stem cells. Next, genome engineering by CRISPR-Cas9 can generate fluorescently labeled fusion proteins such as VSX2-GFP and CRYAA-dTomato to visualize retinal and lens cells, respectively. This double-labeling will aid in the identification of appropriate organoids for long-term cultures. All these tissue-engineering approaches may be needed to establish optimized procedures to produce “dream” lentoid bodies with nearly perfect morphology, including lens sutures as the zones of discontinuity [255].

### 6.2. Advantages of Modeling Human Embryonic Lens Development

The formation of adenohypophyseal, olfactory, and lens placodal cells described above (see Section 4, Figure 1 and Figure 3) is the first direct demonstration of generating anterior pre-placodal ectoderm using a human model, while prior studies employed chicken [36,190,256,257,258], frog [259], and zebrafish cells [260,261]. This model of neural induction and neural plate border was recently re-examined. Interestingly, the chick “pre-border” cells were found comparable with mouse ES cell transcriptomic signatures [262]. Detailed analyses of scRNA-seq analysis of mid-gestation mouse embryos [263,264,265] identified trajectories leading to the individual placodes; nevertheless, the spatial information on cell heterogeneity around the neural plate is still missing. Thus, sorting of pre-placodal cells, visualization of cell migration, and placode aggregation can be modeled using engineered pluripotent stem cells coupled with analysis and direct visualization of individual cells and their clusters.

An impressive number of mouse in vivo models exists that probe cell cycle-coupled differentiation of lens fiber cells using loss-of-function [235,266,267,268,269,270] and gain-of-function [271,272,273,274,275] models. Nevertheless, a number of open questions remain to be addressed that require large-scale protein purifications and analyses of protein–protein complexes and their post-translational modifications at different stages of the cell cycle, e.g., to probe formation of E2F-pRb [275,276], Pax6-pRb complexes [277,278], and interactions between FGF receptors, other membrane proteins, and their intracellular components [53,235,279,280,281,282,283,284,285,286,287,288]. Although dramatic progress with ChIP-seq methods using CUT&RUN, recently implemented in chicken lens studies [226], requires smaller number of cells, genome-wide analysis of lens chromatin occupancy by multiple DNA-binding transcription factors regulating these processes, including p53 [266], Gata3 [289], Hey1 [269], Prox1 [289,290], and Rbpj [291,292,293], can now be performed in parallel with analyses of modified histones and histone variants in chromatin isolated from micro-dissected lenses. For example, loss-of-function of CBP and EP300 histone acetyltransferases disrupts mouse lens placode morphogenesis [294]; however, apart from Pax6, the transcription factors recruiting these enzymes remain mostly unknown in the lens. In addition, we have very limited knowledge on genome localization of nuclear effectors of BMP and FGF signaling in the lens, including transcription factors such as c-Jun, Etv1, Etv4, Etv5, Smad1/5/8, and Smad4. Thus, the requirement for millions of early cells to prepare chromatin can be met using human lentoid bodies. While the in vitro models of lens differentiation described above have limitations and require improvements (see Section 6.1), even current procedures can be scaled up and coupled with cell cycle synchronization and FACS protocols to prepare materials for unbiased multi-omics analyses.

Likewise, determining the individual roles of BPMs, FGFs, their receptors, and nuclear effectors of these signal transduction pathways requires additional studies [28,48,49]. Fluorescent reporter pluripotent stem cells can be engineered using targeted insertions into genes of particular interest, such as transcription and growth factors, to assure that the engineering does not disrupt their normal function. Use of small-molecule inhibitors and/or agonists beyond those listed in Table 3 can be used to modulate their activities and downstream effects on lens cell differentiation [51,52,295,296].

The generation of OFZ is regulated by hypoxia and likely, many other aspects of in vivo lens fiber cell differentiation require hypoxic conditions. It is now feasible to control these levels in the incubator and take full advantage of engineered cell lines to express diverse fluorescent tags and monitor their expression. A recent example is the development of three reporters in single-human iPS cells to tag regulatory genes that control retinal cell differentiation stepwise [297]. Regarding the mechanisms governing OFZ formation, one can simultaneously visualize mitochondria using the mitochondria-targeting sequence “mito” from subunit VIII of human cytochrome c oxidase coupled with GFP [298] and other proteins (mCherry-Mito-7) [299], or the nuclear compartment via core histone protein H2B and lamin B1 fusion proteins. Given the similarities and differences between lens fiber cell de-nucleation [300] and erythrocyte enucleation [301], lens studies can be inspired by pluripotent stem cell cultures that produce human enucleated red blood cells [302].

An increasing number of -omics data exist on the human eye transcriptome [303,304,305,306,307] and proteome [308], including the lens and retinal RNA datasets [146,308] and protein data [309,310,311,312,313]. More data will be available via the NIH-sponsored Human Biomolecular Atlas Program [314]. Any of these data are useful for comparative analyses of lentoid bodies with authentic human lenses. These methods are also useful in the development of standardized quantitative protocols, including integrated epigenetic analyses, to optimize sources of cells for iPS cell procedures [315].

### 6.3. Modeling of Human Early-Onset Congenital Ocular Diseases and Cataracts

There are numerous challenges to understand human congenital eye diseases (Table 4) that affect the lens due to the scarcity of authentic tissues and anatomical and aging differences between humans and mammalian models, such as mice and rats [316]. Cataract remains a leading progressive disease causing blindness worldwide [139]. Many promising drug candidates tested in rodent model diseases fail in human models and clinical testing [317]. Nevertheless, there are many similarities in aging of primate and rodent lenses that demonstrate functional overlaps between different mammalian models [318].

Table 4 shows examples, background, and rationale for systematic studies of transcription factors PAX6, FOXE3, TFAP2A, PITX3, MAF, and HSF4 and their individual mutations causing lens developmental abnormalities and early-onset congenital cataracts. In addition, the microphthalmia-anophthalmia-coloboma (MAC) syndrome [361,362,363,364] is caused by disruption of functionally diverse genes such as OTX2 [364,365], PAX6 [335,336], SOX2 [366], FOXE3 [342], and MAB21L1 [367,368].

The human lens organoid systems are particularly interesting for analyses of congenital cataracts caused by mutations in the main lens structural proteins, e.g., crystallins, MIP/aquaporin 0, gap junction, and cytoskeletal proteins [369,370] (Table 5), as well as key lens-specific transcription factors upstream of crystallin and other lens-specific genes described above (Table 4). The in vitro cataract models discussed here must be directly compared to authentic human cataractous lenses probed by a range of ultramicroscopic [371,372,373] and proteomics [374,375,376,377,378] methods. Generation of mutants primarily for lens/cataract research often finds broader applications to study other eye and central nervous system diseases. For example, mutations in rat Cryba1 affect normal function of RPE and this can be extrapolated into the pathology of human age-related macular degeneration [379,380]. Regarding aging as the major factor in cataract development, current models use genome engineering to induce premature aging through cellular senescence and other mechanisms such as via progerin-induced aging [381], and many organoid cultures are long-lived, such as retinal organoids grown for over 6–9 months [146,147,382]. This approach would be applicable for the modification of lens epithelial cells to disrupt their support of the lens fiber cell compartment. In contrast, a proof-of-principle exists that partial nuclear reprograming by Oct4/Pou5f1, Sox2, and Klf4 of retinal ganglion cells (RGCs) of a mouse model of glaucoma can reverse ageing clocks [383] and can be tested with the lens epithelial cells as well.

Use of human lenses generated from iPS cells to examine molecular mechanisms of cataract formation and their hierarchical structure is already in progress [427]. Treatments of micro-lenses by small drugs indeed affect their transparency and focusing range [137]. Exposure of lentoid bodies to hydrogen peroxide triggers protein aggregation and their opacification [428]. Importantly, two patient-specific iPS cell lines carrying missense and nonsense mutations in CRYBB2 and CRYGD genes (see Table 5), respectively, were reported in 2021 to produce lentoid bodies, followed by detailed morphological and molecular studies [138]. The results nicely demonstrate that lentoid bodies can serve as outstanding models to study protein aggregation and solubility and employ drugs for potential treatments [429,430,431,432].

An important issue is to consider advantages and limitations of using patient-derived iPS vs. isogenic iPS cell lines. Patient-derived iPS cells offer detailed clinical background and phenotypic variability of the inherited mutation. It is necessary to correct the mutation, either heterozygous, homozygous, or compound heterozygous, to produce a pair of cell lines or generate a similar “reference” line from the closest normal relative. If the patient carries a heterozygous mutation, generation of a homozygous mutation represents an additional experimental procedure. These comparative models of two or three iPS cells to generate differentiated cells are instrumental to understand functions of missense and nonsense as well as non-coding mutations such as those affecting splicing, such as in PAX6 [433,434,435], and promoter polymorphism, such as in the αB-crystallin (CRYAB) [436]. To analyze mechanisms of the mutation, it is important to compare it to other mutations that affect the same functional subdomain of the protein. The appropriate patients may not be locally available. In contrast, generation of isogenic iPS cell lines in the laboratory readily produces both hetero- and homo-zygous mutations, and two or more mutations are produced in the same genetic background. In addition, mutations in two or more genes working in the same pathway can be generated. Thus, this approach can expand the repertoire of scientific questions such as complex aging mechanisms to be addressed with these experimental models.

It has been shown earlier that the source of adult cells for reprogramming and generation of iPS cells influences their capacities to be differentiated into specific tissues through their “epigenetic memory” [319,437,438]. The first iPS cells tested for lentoid bodies’ formation were derived from lens epithelial cells collected during age-related cataract surgery of a 56-year-old and three additional individuals, and normal human fibroblasts and H9 ES were used for comparison [127]. Later, normal kidney fibroblasts collected from urine were used [136]. Additional reprograming procedures and primary cells should be tested to determine if there are any differentiation variabilities and produce “standard” protocols for lentoid bodies [139].

Epigenetic regulatory mechanisms are not just important to choose the optimal source of human adult cells and specific reprogramming methods but also serve as important processes underlying cellular differentiation and maintenance of the cell-type memory [439,440,441]. Thus, analyses of DNA methylation and chromatin landscape, pioneered in chicken [227,442] and mouse lenses [443], pave the road for similar studies of human lens cell formation and differentiation using the in vitro procedures. The availability of specific “epigenetic drugs” to pharmacologically target specific enzymes and proteins in charge of individual epigenetic mechanisms provides new tools for future lens research, such as ATP-dependent chromatin remodeling enzymes, DNA methyltransferases and Tet demethylases, histone acetyltransferases and deacetylases, and histone methyltransferases and demethylases. The examples include ATP-dependent chromatin remodeling enzymes Brg1 (Smarca4) [316] and Snf2h (Smarca5) [360], DNA methyltransferases Dnmt1, Dnmt3a, and Dnmt3b [444], histone acetyltransferases EP300 and CBP [295], as well as regulatory subunit Ncoa6 [445] and Znhit1 [446] of the histone methyltransferase MLL3/4 and ATP-dependent SRCAP chromatin remodeling complexes, respectively, which were already examined through loss-of-function studies in the murine lenses.

## 7. Conclusions and Future Research Directions

Despite the current limitations and drawbacks, studies of other organoids have shown that self-organization principles coupled with creative tissue engineering can overcome major obstacles to generate both individual 3D lentoid bodies and primitive eyes comprised of the cornea, lens, and retina [133]. Obviously, the long-term goal of this field is to move from the current micro- and meso-scale into the macroscale order that is needed not only for basic research but for translational applications [5]. Section 6.1, Section 6.2 and Section 6.3 discuss specific examples of questions related to normal human lens development, generation of most advanced lentoid bodies, and various cataract models, respectively. The translational applications include high-throughput testing of candidate anti-cataract drugs as well as testing of drug toxicity within both general and eye-focused clinical trials. There are already studies published in recent years to demonstrate the feasibility of these goals [137,138,139,428,432].

Research on human aging revealed “protective” variants of several genes responsible for longevity [447,448]. It is known that a small fraction of the population does not form age-onset cataracts even in the ninth decade of life [449]. Thus, these individuals would be an excellent source of cells for reprograming and generation of lens organoids to uncover the protective mechanisms using assays such as oxidative stress analyzed by unbiased multi-omics.

In addition to the multi-omics approaches analyzing the chromatin landscape, gene expression, and proteomes in lens cells described above, recent studies (2017–2022) demonstrate the power of metabolomics analyses and different common and gene-specific cataract mechanisms [450,451,452,453,454].

The eye organoids are excellent examples of tissue–tissue coupling, including biomechanical and bioelectrical interactions, vascularization, innervation, host–microbiome interactions, as well as circadian clock entrainment [5,455]. The ocular lens further represents a system with an intricate and compartmentalized microcirculation system and different basic metabolisms, external biomechanical properties, and most importantly, transparency and light refraction [456]. There are challenges of how to mimic these functions; nevertheless, the existing and expanding tool warehouse provides an integrated framework to adapt synergistic engineering modalities to reconstruct all these components in next-generation lentoid bodies, e.g., “bioengineered 3D lenses” or “light-focusing human micro-lenses” [427], and complex eye organoids [5].

One of these challenges relates to the precise 3D lens shape and its control via a combination of processes governing lens growth, optic, and mechanical stretching to generate a gradient refractive index underlying physiological lens function [457] that may not be possible without complex tissue engineering to mimic functionally active lens ciliary muscles and zonules [458,459].

To illustrate the range of options in tissue bioengineering, these recent pioneering studies within the eye research require further attention. Human iPS cell-generated trabecular meshwork cells, normally of mesenchymal origin, cultured on epoxy-based biologically, fully compatible SU-8 microfabricated scaffolds, show expected morphology, ECM deposition, and physiological responses to dexamethasone treatment [460,461]. Other examples are development of the Bruch’s membrane-mimetic substrate to grow RPE cells using 3D printing technology to model various retinal degenerations [462], generation of clinical-grade patches of human RPE cells from iPS cells using biodegradable scaffolds [152], and use of low-fibril density thin-collagen vitrigels as 3D scaffolds [463] to improve maturation of human RPE [464]. Finally, a recent study established lacrimal gland organoids from minced murine and diagnostic human lacrimal glands treated by a combination of growth factors and small-molecule drugs [465]. These experiments demonstrate additional opportunities of how to employ primary cells in the eye organoid research.

Organ-on-a-chip technology applied to vision research remains in its infancy [466]. This biomedical engineering research platform is a multi-channel 3D microfluidic cell culture system to grow artificial organs of various complexity to mimic as many physiological parameters as possible of the normal organ, its vasculature, nutritional regimens, growth factor flow, and mechanical forces [467,468], with some features already applied to retinal [469,470] and corneal [471] research. A general challenge of these studies is the sustainability of the long-term cultures. We estimate that the bioengineered 3D lenses should reach sizes within the 2–10 mm scale within 12–16 weeks.

Taken together, the research on lens and eye organoids is poised for new breakthrough discoveries through collaborations between lens and vision researchers with experts in tissue engineering and bioprinting, individual multi-omics methods, genome engineering, high-throughput methods of drug screening, and other related disciplines. Experimental challenges remain in the ambitious goal to produce in vitro-generated macroscale human lenses with as close as possible morphology and size compared to authentic human embryonic and adult lenses, as current lentoid bodies and micro-lenses only partially meet these requirements. In parallel, efforts to generate 3D human eyes are not far away given the rapid pace of earlier transformative discoveries summarized in this review.

## Figures and Tables

**Figure 1 cells-11-03516-f001:**
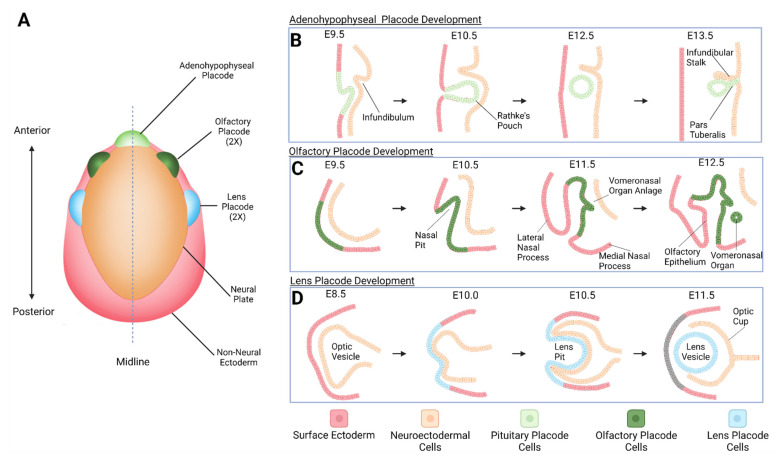
Formation of three anterior cranial placodes during mid-stage mouse embryogenesis. (**A**) Highly schematic visualization of future location of adenohypophyseal, olfactory, and lens placodes relative to the neural tube and its anterior/posterior patterning at the E8.0 mouse embryo. Regarding the eye and lens formation, note that the eye field is already formed within the anterior subregion of the neural plate [30] (not shown), and undergoes its symmetric division, later forming a symmetric pair of optic vesicles (see panel D). (**B**) Adenohypophyseal (pituitary) gland development from E9.5 via Rathke’s pouch [31] to E13.5, forming the infundibular stalk and the pars tuberalis [32]. (**C**) Olfactory placode development from E9.5 to E12.5, forming the vomeronasal organ [33]. (**D**) Lens placode development from E8.5 to E11.5, forming the lens vesicle [34]. The prospective corneal epithelium formed after the separation of the lens vesicle from the surface ectoderm is highlighted in gray. Note that for simplification, neural crest cells including periocular mesenchymal cells are not shown, but they are generally located in the space between the neuroectoderm and surface ectoderm. For additional details, see https://syllabus.med.unc.edu/ for ultrastructural images of mouse eye development between these stages.

**Figure 2 cells-11-03516-f002:**
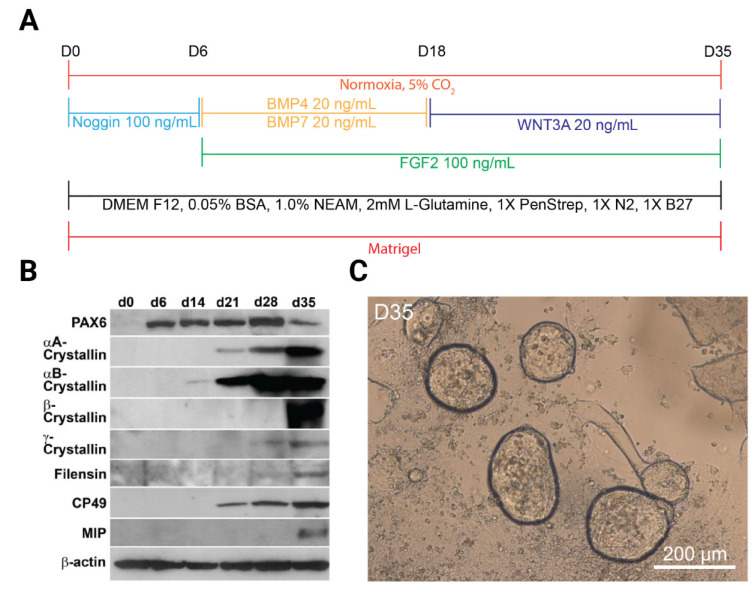
Summary of three-stage procedure to generate human lentoid bodies. (**A**) Lentoid bodies were differentiated over a 35-day period on Matrigel using specific concentrations of Noggin, BMP4, BMP7, FGF2, and WNT3A at the time points shown. Additional components and their concentrations of the basal medium are also shown [125]. (**B**) Western blot analysis shows expression of key lens markers PAX6, αA-, αB-, β-, and γ-crystallins, filensin (BFSP1), CP49 (BFSP2), and MIP (AQP0). Expression of β-actin was used as a loading control [125]. (**C**) Brightfield image of lentoid bodies produced at day 35 of differentiation.

**Figure 3 cells-11-03516-f003:**
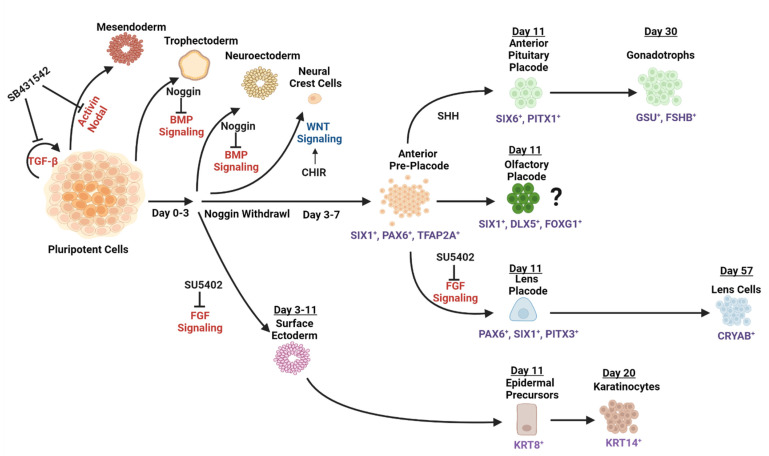
A summary model of the generation of anterior placodes from hPSCs using the dSMADi-based procedures. The diagram illustrates multiple cell fate decisions of the pluripotent stem cells, including those following Noggin- and SB431542-mediated phase of inhibition and TGF-β, Activin, and Nodal signaling inhibition [139]. Formation of anterior pre-placodal cell fate commitment can be subsequently induced through BMP activation. In contrast, epidermal precursors can be generated through surface ectoderm induction mediated by FGF inhibition. The anterior pre-placodal cells can generate anterior pituitary placodal cells through activation of SHH signaling to produce gonadotrops at day 30 of the cultures [139]. Note that these cells treated subsequently by FGF8, BMP2, and both factors together generate hormone-producing cells analyzed at day 60 [140]. The olfactory placodal cell fate pathway analysis was limited to identification of a few cell markers, SIX1, DLX5, and FOXG1, and is marked (?) accordingly. Addition of BMP4 between days 7 and 11 increased the expression of lens lineage-specific transcription factors PAX6 and PITX3. Lens placodal cells were enriched through inhibition of FGF signaling by SU5402 at day 7, followed by analyses of lens placode markers PAX6, SIX3, and PITX3 and subsequent formation of αB-crystallin-expressing cells analyzed at day 57 [139].

**Table 1 cells-11-03516-t001:** A representative list of human ES/iPS differentiation methods to generate the entire spectrum of ocular cell types.

Tissue(s)	Cells	Differentiation Procedures, Outcomes, and References	Growth Factors/Small Molecules *
Whole eye-like structures	mESCs	PA6 feeder cell system, self-organizing ocular-like structures including RPE, lens, neural retina [105]	**None**
Whole eye-like structures	hiPSCs	Self-formed ectodermal autonomous multi-zone (SEAM) protocol using highly specific LN211E8 laminin matrix, self-organizing ocular-like structures including RPE, lens, neural retina, and cornea [134]	**Noggin, SB431542, FGF2,** KGF, Y27632
Whole eye-like structures	hiPSCs	2-step method to obtain self-organized multizone ocular progenitor cells (mzOPCs) on Matrigel → isolated 3D-suspension culture producing RPE, neuroretina, cornea-like structures with less frequent clusters of lens cells [135]	**Noggin, DKK1** **IGF1, FGF2** followed by IGF1 and ATRA, completed by IGF1 and ATRA (cornea) or IGF1 and T3 (photoreceptors and RPE)
Lens progenitor cells and lentoid bodies/micro-lenses	hESCs, hiPSCs	3-stage protocol on Matrigel, self-organizing lens progenitors and lentoid bodies [126], “fried egg” modification [136], ROR1-based cell sorting [137], and large-scale use [138]	**Noggin, BMP4, BMP7, FGF2, WNT3A**
Lens and other placodes	hESCs hiPSCs	Dual-SMAD inhibition (dSMADi) using Matrigel, self-organizing anterior pituitary, trigeminal, and lens placodes [140,141,142]	**SB431542, Noggin, SHH, SU5402, FGF2**, FGF8, BMP2, BMP4
Lens and other placodes, lentoid bodies	hESCs	Matrigel-grown cells in insulin-transferrin-selenium (ITS) serum-free medium, HGFR (c-MET), and CD44 sorting to isolate proliferative lens epithelium populations [143]	**None**
Pre-placodal ectoderm, lens placodes, and early lens spheres	hESCs	Adherent culture on Matrigel differentiated into pre-placodal progenitors able to produce lens placode, pituitary, and other precursors [144]	**Noggin, SB431542, BMP4, bovine vitreous humor**, FGF2, 6-BIO (GSK3β inhibitor), dorsomorphin (AMPK inhibitor), **LDN193189, cyclopamine** (SHH inhibitor), RA, purmorphamine (Smo receptor agonist)
Lens progenitor cells and lentoid bodies	mESCs hESCs	Transfection and transduction of *Pax6* and *Six3* induced differentiation of lens progenitor cells grown on mouse embryonic fibroblast feeder cells [129]	**None**
Lens progenitor cells and lentoid bodies	miPSCs	Engineered miPSCs with Cryaa/tdTomato reporter were grown on NTERA-2 and P19 feeder cells [131]	**None**
RPCs and optic cups	mESCs hESCs	Self-organizing optic cups, retinal progenitors, neural retina in Matrigel suspension culture [110,111]	Y27632, IWR1e, SAG, CHIR99021
RPCs (floating retinas)	hESCs	Retinal organoids formed from floating cysts suspended in Matrigel [145,146]	Y27632
RPC and photoreceptors	hiPSCs	Self-organizing suspension culture system that generates retinal cups with all major retinal cell types [147]	RA
Neural retina and RPE	hESCs	Generation of retinal organoids with increased efficiency using IGF-1 signaling [148]	IGF1
RGCs	hESCs	Generation of transplantable retinal ganglion cells in Matrigel culture [149,150]	Forskolin, IDE2, DAPT, dorsomorphin, nicotinamide
Cone photoreceptors	hESCs	Generation of M- and S-opsin-positive cones with outer segment-like structures in Matrigel culture [151]	Noggin, DKK1, IGF1, DAND5/COCO, FGF2
RPE	hiPSCs	Efficient generation of RPE cells by chemically defined conditions using a triphasic protocol [152]	LDN193189, SB431452, CKI-7, IGF1
RPE	hiPSCs	RPE cells generated by an unbiased differentiation approach using high-throughput compound screen and hiPSC-based RPE reporter assay [153]	Chetomin, nicotinamide
Corneal epithelium	hiPSCs	Biphasic culture method using CnT-30 media that transitions from suspension to adherent culture generating corneal epithelium [154]	SB505124, IWP-2, FGF2
Corneal epithelium	hiPSCs	4-stage protocol transitioning from suspension to adherent culture using supplemented hormonal epithelial medium and small drug ROCK inhibitor, Y27632 [155]	Y27632
Corneal epithelium	hiPSCs	Biphasic culture method using CnT-30 media that transitions from Matrigel- to Collagen-IV-coated plates [156]	BMP4, Trans-RA, EGF, SB505124, IWP-2, LDN193189, Y27632, SB431542
Corneal endothelium	hiPSCs	Corneal endothelium cells generated using TGF-β inhibitor and GSK-3 inhibitor cultured on laminin-521 [157]	LDN193189, SB431542, CHIR99021, RA
Corneal limbus	hiPSCs	Corneal limbal progenitors derived using Panserin 801 medium supplemented with BMP4 cultured on fibronectin and laminin [154,158]	BMP4
Trabecular meshwork	miPSCs	Induction of trabecular meshwork (TM) cells using miPSCs cocultured with human trabecular meshwork (TM) cells [159]	None
Trabecular meshwork	miPSCs	Induction of TM cells using miPSCs using media conditioned by human TM cells [160]	None
Trabecular meshwork	miPSCs	Two-step induction of TM cells through verification of neural crest cell intermediary stage grown in TM cell-conditioned media [161]	Y27632
Periocular mesenchyme	hESCs	Periocular mesenchymal cells generated from hESCs using TGF-β and WNT signaling inhibition [162]	SB431542, ATRA, IWP-2

* Growth factors/small molecules listed here were used towards the lens cells and lentoid bodies (**bold font**, see Table 3 for details). Other factors evaluated (alone or in combinations) in the systems for other purposes are shown to illustrate the range and spectrum of currently known possible treatments of eye organoids. Abbreviations/codes: All-trans retinoic acid, ATRA; GSK3 inhibitor, CHIR99021; CKI-7, casein kinase 1 inhibitor; DAN domain BMP antagonist family member 5 DAND5 (COCO), γ-secretase inhibitor, DAPT; epidermal growth factor, EGF; small-molecule inhibitor of Wnt signaling, IWP-2; Wnt/β-catenin signaling inhibitor, IWR1e; fibroblast growth factor 7, FGF7 (also known as keratinocyte growth factor, KGF); Rho kinase inhibitor, Y27632; SAG, hedgehog pathway activator of SMO receptor; GSK3α/β inhibitor, 6-BIO; inhibitor of TGF-β receptors, SB505124; triiodothyronine, T3.

**Table 2 cells-11-03516-t002:** Marker genes expressed during the formation of lens and olfactory placodes.

Region	Stage	Genes Expressed [33,163,164,165]
aPPR	E8–8.5	Dlx2, Dlx5, Foxg1, Otx2, Six1, Six4
Lens/olfactory	E8.5–8.75	Aldh1a1, Aldh1a3, Gata3, Pax6, Six1, Six3
Lens	E9–E9.5	Aldh1a1, Aldh1a3, Cryab, Eno1, FoxE3, Gata3, Has2, Lama1, Mab21l1, Maf, Mafb, Prox1, ROR1, Six3, Sox2
Olfactory	E9–E9.5	Aldh1a3, Dlx6, Dmrt4, Emx2, Ebf2, Fgf8, Foxd4, Ngn1, Pax6, Six1, Six4, Sox2, Sox3

**Table 3 cells-11-03516-t003:** Growth factors and small-molecule drugs used in lens progenitor cell and lentoid bodies’ formation.

Growth Factor or Inhibitor	Structure, Function, and References	Application in Generation of Lens Progenitor Cells and Lentoid Bodies
**Noggin**	BMP inhibitor (232 aa, 25.7 kDa), adopts a cysteine-knot fold tertiary structure to form a homodimer that binds BMP homodimer to form a ring-like structure that prevents BMPs to interact with their receptors in a reversible manner [171], binds BMP2, 4–7, 13, and 14	Required for neuroectoderm induction [125,139,165]
**FGF2**	Heparin-binding protein family (288 aa, 30.7 kDa), globular structure comprised of β-strands, α-helices, and two turns, binds receptors FGFR1–4 [172]	Formation of lens progenitors [125], lentoid bodies’ differentiation [135,136]
**BMP4**	SMAD signaling activator (408 aa, 46.5 kDa), homodimers bind type 1 receptors BMPRIA and type 2 BMPR2. Once these three molecules are bound together, BMPR2 phosphorylates and activates BMPR1A [48]	Epidermal induction, averts neural cell fate; formation of lens progenitors [125] and pre-placodal cells [139]
**BMP7**	SMAD signaling activator (431 aa, 49.3 kDa), homodimers bind type 1 ACVR1 (ALK2) and BMPRIB (ALK6), and type 2 receptor ACVR2A [173,174]	Used to generate lens progenitor cells and lentoid bodies [125]
**WNT3A**	β-catenin-dependent Wnt signaling activator (352 aa, 39.4 kDa), binds receptors Frizzled, LRP5, and LRP6 signaling complex [175]	Improves FGF2-dependent lentoid body formation [125]
**DKK1**	β-catenin-dependent Wnt inhibitor (266 aa, 28.7 kDa), binds via C-terminal Cys-rich domain co-receptor LRP6 via a tandem of β-propeller regions [176]	Formation of the presumptive lens ectoderm in SEAM [133]
**IGF-1**	Secreted insulin-like growth hormone (195 aa, 21.8 kDa), binds to the α-subunit of IGF1R receptor and integrin complexes ITGAV:ITGB3 and ITGA6:ITGB4 [177]	Promotes formation of 3D primitive ocular-like structures that contain primitive lenses [178]
**SB431542**	SMAD signaling inhibitor drug derived from benzamide (MW: 384), inhibits type 1 receptors TGFBRI (ALK5), ACVR1B (ALK4), and ACVR1C (ALK7) [179]	Neuroectoderm induction (dSMADi) [139,165]
**SU5402**	Pyrrole-3-propanoic acid derivative (MW: 296), inhibits tyrosine kinase receptors VEGFR2, FGFR1, and PDGFRβ [180]	Disrupts placode induction and triggers epidermal cell fate and later promotes lens cell formation [139]
**LDN193189**	Pyrazolol and pyrimidine derivative of quinoline (MW: 406.5), inhibits type 1 ACVR1 (ALK2), BMPR1A (ALK3), and BMPR1B (ALK6) receptors [181]	“Transitional” BMP inhibition by this drug favors expression of PAX6, while BMP4 added later strongly induces lens placodal cells [143]
**Cyclopamine**	Steroidal alkaloid (MW: 411.6) with a high affinity for cell surface membrane protein Smoothened (SMO), a G-protein-coupled receptor (class F) which works downstream of Sonic hedgehog (SHH) 12-pass transmembrane protein receptor Patched (PTC) [182,183,184]	Inhibition of SHH signaling by this drug boosts FOXE3 and CRYAA expression [143]

**Table 4 cells-11-03516-t004:** Examples of model transcription factors (TFs) and their roles in lens development.

Name	Structure and Function	Representative Human Protein Mutations for Future Studies
**PAX6**	Paired domain (4–132 aa) and homeodomain (210–269 aa) TF (422/436 aa, 46/48 kDa). Regulates all stages of lens morphogenesis through its expression in lens precursor, progenitor, lens epithelium, and lens fiber cells [21,23,319,320,321]	Nonsense mutations: R203X, R240X, R317X, and R353X; missense mutations: G18W, R26G, G64V, R128C, and R242T [322,323,324,325,326,327,328,329,330,331,332]
**TFAP2A**	Non-canonical AP-2-type DNA-binding helix-span-helix and dimerization domain (280–410 aa) TF (437 aa, 52 kDa). Regulates separation of the lens vesicle from the surface ectoderm and maintenance of lens epithelium [171,333,334]	Missense mutation: H384Y [335]
**FOXE3**	Forkhead domain (71–165 aa) TF (319 aa, 33 kDa). Required for lens epithelium morphogenesis and lens formation [336,337]	Missense mutations: A96T, R120P, and T124M [338]
**PITX3**	Homeodomain (62–121 aa) TF (302 aa, 31.8 kDa). Deletion in mouse Pitx3 promoter causes aphakia (absence of the lens) via degeneration of the lens vesicle [339,340,341]	Nonsense C240X [342], missense S13N and G219fs mutations [343]. Frameshift Ser192Alafs* 117 mutation [344]
**MAF**	Basic motif (288–313 aa) and leucine zipper (316–337 aa) bZIP TF (373 aa, 38.5 kDa), binds to DNA as homo- or hetero-dimers. Required for lens fiber cell elongation and crystallin gene expression [295,345,346,347,348,349]	Missense mutations: G273N, R294W, C305W [350], R288P [351], and S270T [352]
**HSF4**	Helix-turn-helix DNA-binding domain (17–121 aa) and two leucine zipper oligomerization domains to form a triple-coiled-coil trimeric structure on DNA of this TF (492 aa, 53 kDa). Abnormal lens fiber cell differentiation and disrupted de-nucleation [353], binding to αB- [354] and γ-crystallin promoters [353], regulation of DNase IIβ expression [355,356], and other lens-specific genes [357]	Missense mutations: K64E [358], R116H [359], and Homozygous splice mutation of intron 12 (c.1327 + 4A > G) [360]

**Table 5 cells-11-03516-t005:** Examples of mutated cataract genes encoding lens structural proteins and their current/future models in lentoid bodies.

Gene/Protein	Structure and Function	Representative Human Protein Mutations in Ongoing and Future Studies
CRYAA/HSPB4	αA-crystallin (173 aa, 19.9 kDa). Small heat-shock-like protein (52–164 aa), molecular chaperone-like activities. Both α-crystallins form heterogeneous multimeric assemblies between 16 to over 50 subunits and represent about 35% of lens soluble proteins [384,385]	R49C [386] and R116C [387] missense mutations, single amino acid deletion p.117delR [388], 174Sext * 41 and 174Gext * 41 mutations in the stop codon following 41 aa extension [389]
CRYAB/HSPB5	αB-crystallin (175 aa, 20.2 kDa). Small heat-shock-like protein (56–164 aa), molecular chaperone-like activities (see above for CRYAA).	G154S, R157H, and A171T missense mutations cause cataract and heart defects [390]
CRYBA1	βA1/A3-crystallin (215 and 198 aa, 25.2 and 23 kDa). The β/γ-crystallins originated from gene duplication of an ancestral gene encoding two domain proteins. Each domain consists of four β-strands, forming a “Greek key” structural motif [391,392,393]	Splice mutation c.215 + 1G > A [394], G91del mutation of 3 bps [395]
CRYBB1	βB1-crystallin (252 aa, 28 kDa) [392,393,394]	Missense S129R mutation [396]
CRYBB2	βB2-crystallin (205 aa, 23.4 kDa) [392,393,394]	Missense P24T/iPS cells [138] and nonsense Q155X [397]
CRYGC	γC-crystallin (174 aa, 20.9 kDa) [392,393,394]	G129C missense mutation causing nuclear cataract [398]
CRYGD	γD-crystallin (174 aa, 20.8 kDa) [392,393,394]	R36C mutation causes microcrystals within the lens [399], W43R [400], and Q155X/iPS cells [138]
MIP/AQP0	Lens fiber major intrinsic protein/water channel (aquaporin 0), six transmembrane passes, C-terminal region binds BFSP1 (263 aa, 28.1 kDa) [401,402]	Missense E134G and T138R [403,404] mutations.
LIM2/MP19	Lens-specific membrane protein (173 aa, 19.7 kDa) involved in organization of cell junctions and receptor for calmodulin, four transmembrane domains [405,406]	Missense G78D [407] and R130C [408,409] mutations.
GJA8/connexin 50	Gap junction protein alpha 8 (433 aa, 48.2 kDa) in lens fibers. N-terminal intramembrane domain followed by three transmembrane domains and C-terminal extracellular domain [410,411,412].	Missense mutations in second transmembrane domain P88S [413] and P88Q [414].
GJA3/connexin 43	Gap junction protein alpha 3 (435 aa, 47.4 kDa); four N-terminal transmembrane domains, four types of dimers: homomeric, heterotypic homomeric, heterotypic heteromeric, and heteromeric [415,416].	P32L missense mutation in the first transmembrane domain [417], T19M [418].
EPHA2	EPH-receptor A2 single-pass transmembrane protein with C-terminal tyrosine kinase domain, ephrin ligand binding, cysteine-rich EGF-like domain, and two fibronectin III-type repeats binding in the N-terminal extracellular domain (976 aa, 130 kDa) [419].	Missense G668D in kinase domain [420], R175C in N-terminal ligand-binding domain [421], splice-site mutation c.2826 − 9G > A [422].
BFSP1/filensin/CP115	Lens-specific intermediate filament protein (665 aa, 74.5 kDa). Head (1–40 aa), intermediate filament rod (40–320 aa) [59].	Splicing mutation: c.625 + 3A > G [423]; iPS cells with double heterozygous mutations Glu269Lys in BFSP1 and L125P in RHO gene [424].
BFSP2/phakinin/CP49	Lens-specific intermediate filament protein (415 aa, 46 kDa). Head (2–114 aa), intermediate filament rod (104–415 aa) [59].	ΔE233 [425].
TDRD7	RNA-binding protein (1098 aa, 123.5 kDa), two Tudor domains (513–570 and 703–760 aa) [426].	V618del in-frame deletion of 3 bps [426].

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
