# Peer review of "Generation of Lens Progenitor Cells and Lentoid Bodies from Pluripotent Stem Cells: Novel Tools for Human Lens Development and Ocular Disease Etiology"

_cells, 2022, doi:10.3390/cells11213516_

Round 1

Reviewer 1 Report

a very nice review describing the Generation of lens progenitor cells and lentoid bodies from 2 pluripotent stem cells. 

here are some of minor concerns:

1- the section should be tilted and subtitled

2-The conclusion should be short and informative

3-the authors should upstate the reference to recent advance in the felid

4-The figures are clear

5-overall, the article is good but need more cutting-edge paragraphs to show up the influence of literatures impact. 

Author Response

Referee: 1

A very nice review describing the Generation of lens progenitor cells and lentoid bodies from 2 pluripotent stem cells. Here are some of minor concerns:

1-the section should be tilted and subtitled.

Response: We generated new subsections 4.1 - 4.3. Other sections 2., 3. and 7 do not require any subsections as they are much shorter compared to sections 5. and 6, each with three subsection already in the original manuscript.

2-The conclusion should be short and informative.                                                              

Response: This comment most likely relates to Section 7. Conclusions and future research directions are now comprised of eight paragraphs, including a new section based on Reviewer 2 suggestion.

3-the authors should upstate the reference to recent advance in the felid.                         

Response: As the references are numerical, we included years of research and/or publications all across the manuscript to highlight what was accomplished 20 years ago, between 2010-2018 and in past 5 years. Most recent studies from 2017-2022 are now also mentioned at the end of the first paragraph of the final section 7.

4-The figures are clear.                                                                                                  

Response: N/A

5-overall, the article is good but need more cutting-edge paragraphs to show up the influence of literatures impact.                                                                                                             

Response: Multiple edits were done across the main text in response to this suggestion.

Reviewer 2 Report

I applaud the authors for the comprehensive review and discussion about this important topic in the lens community and beyond. I enjoyed reading the article and really have no major comments. 

Lens epithelial cell differentiation is not equally occurring in the development and post-natal growth period. Lens epithelial cells at the germinative zone but not the central zone are highly active in cell proliferation and differentiation. It is a coordinative process supported by the local environment, such as EGF and other fiber cell differentiation factors near the equator region. Lens shape is a result of these unique phenomena. It would be great if the authors could discuss the approaches and challenges in generating "normal" shape lenses.  

Author Response

Referee 2:

I applaud the authors for the comprehensive review and discussion about this important topic in the lens community and beyond. I enjoyed reading the article and really have no major comments. 

Lens epithelial cell differentiation is not equally occurring in the development and post-natal growth period. Lens epithelial cells at the germinative zone but not the central zone are highly active in cell proliferation and differentiation. It is a coordinative process supported by the local environment, such as EGF and other fiber cell differentiation factors near the equator region. Lens shape is a result of these unique phenomena. It would be great if the authors could discuss the approaches and challenges in generating "normal" shape lenses. 

Response: We thank the reviewer for raising this issue. We inserted a new paragraph in Section 7, including three new references. We also comment on PDGF and PI3K signaling to promote lens growth and OFZ formation, including four new references.